# Don't Judge Before You CLIP:
# A Unified Approach for Perceptual Tasks

**Amit Zalcher**[*]                                                        *amit.zalcher@weizmann.ac.il*
*Department of Computer Science and Applied Mathematics*
*The Weizmann Institute of Science*

**Navve Wasserman**[*]                                              *navve.wasserman@weizmann.ac.il*
*Department of Computer Science and Applied Mathematics*
*The Weizmann Institute of Science*

**Roman Beliy**
*Department of Computer Science and Applied Mathematics*
*The Weizmann Institute of Science*

**Oliver Heinimann**
*Department of Computer Science and Applied Mathematics*
*The Weizmann Institute of Science*

**Michal Irani**
*Department of Computer Science and Applied Mathematics*
*The Weizmann Institute of Science*

**Reviewed on OpenReview:** *https://openreview.net/forum?id=uvQTYi6kbu*

## Abstract

Visual perceptual tasks aim to predict human judgment of images (e.g., emotions invoked by images, image quality assessment). Unlike objective tasks such as object/scene recognition, perceptual tasks rely on subjective human assessments, making their data-labeling difficult. The scarcity of such human-annotated data results in small datasets leading to poor generalization. Typically, specialized models were designed for each perceptual task, tailored to its unique characteristics and its own training dataset. We propose an identical architectural framework for solving multiple different perceptual tasks leveraging CLIP as a prior. Our approach is based on recent cognitive findings which indicate that CLIP correlates well with human judgment. While CLIP was explicitly trained to align images and text, it implicitly also learned human inclinations. We attribute this to the inclusion of human-written image captions in CLIP's training data, which contain not only factual image descriptions, but inevitably also human sentiments and emotions. This makes CLIP a particularly strong prior for perceptual tasks. Accordingly, we suggest that minimal adaptation of CLIP suffices for solving a variety of perceptual tasks. Our simple unified framework employs a lightweight adaptation to fine-tune CLIP to each task, without requiring any task-specific architectural changes. We evaluate our approach on three tasks: (i) Image Memorability Prediction, (ii) No-reference Image Quality Assessment, and (iii) Visual Emotion Analysis. Our model achieves state-of-the-art results on all three tasks, while demonstrating improved generalization across different datasets. See our **Project page** for demos and models.

---

[*]Equal contribution

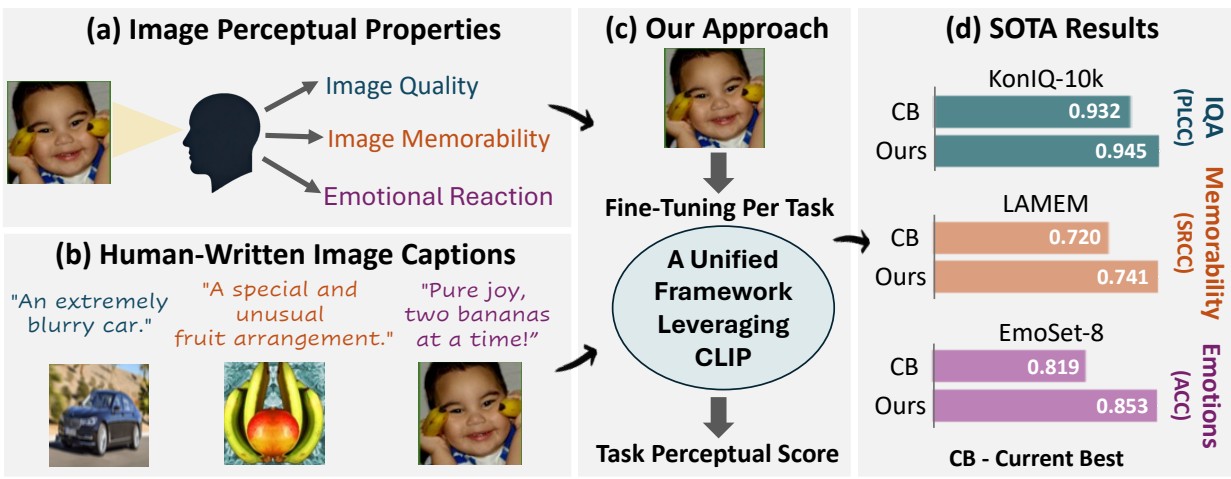

Figure 1: **Our Framework:** *(a) Perceptual tasks rely on subjective human judgment. (b) Illustration of CLIP's training samples, which includes human-written captions. These human-generated annotations contain not only factual image descriptions, but inevitably also human sentiments, preferences and emotions. This suggests that CLIP can serve as a prior for perceptual tasks. (c) Our approach leverages CLIP's prior knowledge to address multiple perceptual tasks with minimal task-specific adaptation (d) We achieve state-of-the-art performance across three distinct perceptual tasks. CB refers to the current best method in each task (see Tables 1,3,4 for numerical scores).*

## 1 Introduction

Visual perceptual tasks have been a long-standing research area (Marr, 2010; Leeds et al., 2013), incorporating both psychological aspects (Gibson, 2014; Lu et al., 2011; Shemyakina et al., 2024) and practical applications like education (Wang, 2021; Duivenboden, 2023) and advertising (Pieters et al., 2010; Bi et al., 2024; Sample et al., 2020; Negm & Tantawi, 2015). These include predicting human-perceived image quality (Khosla et al., 2015; Ma et al., 2017; Zhao et al., 2023), emotional responses to images (Peng et al., 2016; Yang et al., 2021a), and image memorability (Isola et al., 2011b; Squalli-Houssaini et al., 2018). Unlike objective tasks such as object or scene recognition, perceptual tasks rely on *subjective* human assessments, making its data-labeling difficult. First, capturing genuine human reactions requires carefully designed experiments, such as using precise survey questions to distinguish subtle emotional states evoked by images. Additionally, variability in perception of different people necessitates collecting data from multiple participants to ensure robust and reliable results (Kret & De Gelder, 2012; Isola et al., 2011b; Fang et al., 2020). The scarcity of such human-annotated data results in small datasets which restricts the capabilities and generalization of predictive models.

Typically, specialized models were designed for each perceptual task, tailored to its unique characteristics and its own training dataset. For instance, emotion recognition models (Xu et al., 2022; 2024b) often integrate psychological insights, while image quality assessment models (Xu et al., 2024a; Qin et al., 2023) focus on multi-level image features. However, fully understanding the specific characteristics of human perception for each perceptual task is an ongoing research and remains debatable (Kramer et al., 2023; Chandler, 2013; Zhang et al., 2024; Wang et al., 2023a).

Since human-labeled data is limited, a strong *perceptual* prior is needed. In this paper we propose a unified approach, named **PerceptCLIP**, for solving multiple different perceptual tasks using identical architectural frameworks. Our approach is based on recent cognitive findings (Shoham et al., 2024) which indicate that CLIP correlates well with human judgment. While CLIP was explicitly trained to align images and text, it has also implicitly learned human preferences. We attribute this to the nature of CLIP's training data, which includes human-written image captions that consist not only factual image descriptions, but inevitably also *subjective* aspects such as human sentiments, preferences and emotions. This makes CLIP a particularly strong prior for *perceptual tasks*. Accordingly, we suggest that *minimal* adaptation of CLIP, rather than extensive fine-tuning, is sufficient to achieve strong performance on perceptual tasks.

A few previous methods employed CLIP for specific perceptual tasks. However, some *fully* fine-tuned most of its components (Zhang et al., 2023; Martín-Fernández et al., 2023), thus erasing its strong pre-trained knowledge and risking dataset-overfitting. Others relied on tuning only CLIP's text-prompts (Deng et al., 2022; 2024; Wang et al., 2023b; Xu et al., 2024b), which is too restrictive, preventing proper adaptation of relevant layers. *Our approach strikes a balance between preserving CLIP's strong ability to capture diverse perceptual attributes, while allowing sufficient flexibility for task-specific adaptation.*

Specifically, our simple unified architectural framework employs a lightweight adaptation to fine-tune CLIP to each task, without requiring any task-specific architectural changes. We apply LoRA (Low-Rank Adaptation) (Hu et al., 2021) on CLIP visual encoder, selectively *fine-tuning only the attention layers*, followed by an MLP head. This approach preserves CLIP's perceptual priors while enabling task-specific refinement.

We evaluate our approach on three tasks: (i) Image Memorability Prediction, (ii) No-reference Image Quality Assessment, and (iii) Visual Emotion Analysis. Our model achieves state-of-the-art results on all three tasks, while demonstrating improved generalization across different datasets. Our results show that CLIP already holds rich perceptual knowledge and, with efficient tuning, surpasses previous task-specific methods without extensive manual adjustments or domain expertise. Furthermore, we demonstrate the benefits of joint training on multiple datasets of the same task, by using a different MLP head per dataset, with a joint finetuned CLIP backbone for all datasets. This significantly enhances model performance on small datasets.

**Our contributions are as follows:**

- We propose a *simple unified framework* that leverages CLIP's pre-trained knowledge, enabling effective adaptation to multiple perceptual tasks *and eliminating the need for task-specific architectural adjustments.*

- Our models achieve *state-of-the-art performance across 3 different perceptual tasks*: image memorability prediction, no-reference image quality assessment, and visual emotion analysis, showcasing both superior results, as well as better generalization across different datasets.

- We demonstrate that joint training on multiple different datasets of the same task, leads to enhanced model performance on small datasets.

- We highlight that strong, reproducible baselines using simple framework can outperform complex, task-specific architectures, setting a solid foundation for future progress.

## 2 Related Work

**No-Reference Image Quality Assessment (IQA).** No-reference IQA involves predicting the perceptual quality of an image (according to human judgment) without a reference image. Early methods predominantly used CNN-based architectures (Kang et al., 2014; Ma et al., 2017; Su et al., 2020; Network; Saha et al., 2023) or hybrid CNN-transformer models (You & Korhonen, 2021; Xu et al., 2024a). These approaches aimed to capture both local and global image features, essential for accurate quality assessment (Qin et al., 2023). More recent transformer-based models have demonstrated further improvement (Qin et al., 2023; Ke et al., 2021). A notable advancement in IQA has been the integration of large-scale pretrained models (Zhang et al., 2023; Xu et al., 2024a; Wang et al., 2023b). While CLIP has been utilized as a pretrained backbone (Zhang et al., 2023; Wang et al., 2023b), existing approaches either fully fine-tune it or optimize only text prompts, failing to effectively adapt CLIP to the task and fully leverage its strong prior. The current state-of-the-art (Xu et al., 2024a) uses both a pretrained ViT and ResNet trained on ImageNet, adapting them to the IQA task. Our method surpasses these approaches while having significantly fewer trainable parameters.

**Image Memorability Prediction.** Early works (Isola et al., 2011a;b) demonstrated that image memorability (i.e., the likelihood of an image to be remembered) is an inherent property of an image, influenced primarily by its content and structure. Those initial approaches relied on global image descriptors and color histograms, later evolving to leverage features extracted from fine-tuned CNNs (Khosla et al., 2015),

which significantly improved prediction performance. Subsequent methods (Squalli-Houssaini et al., 2018; Leonardi et al., 2019) incorporated additional semantic features from image captioning systems or soft attention mechanisms. Residual networks (ResNets) further enhanced memorability estimation, with studies like Fajtl et al. (2018) integrating ResNet50 and LSTM for regression. A recent work (Hagen & Espeseth, 2023) explored using Vision Transformer (ViT) (Dosovitskiy et al., 2020), which performed well but still fell short of Squalli-Houssaini et al. (2018). Our approach achieves superior performance compared to the above methods leveraging CLIP perceptual prior.

**Visual Emotion Analysis.** The problem of assessing the emotional reaction and sentiment of people towards images has evolved through various approaches. Early works (Yang et al., 2018; Rao et al., 2019) used pre-trained CNN backbones for feature extraction, while graph-based methods (Yang et al., 2021a; Wu et al., 2021; Yang et al., 2021b) emphasized object-graph relationships and emotion-enhanced features, highlighting the importance of contextual interactions. Xu et al. (2022) introduced hierarchical emotion modeling, leveraging psychological theories for improved emotion representation. CLIP has recently been explored for emotion recognition (Deng et al., 2022; 2024; Xu et al., 2024b; Yao et al., 2024), mainly through prompt-based learning while some further use insights from psychological research. Our method demonstrates improved results, without relying on task-specific knowledge.

Unlike previous methods that rely on task-specific architectures, we use a single architecture across tasks, eliminating the need for extensive modifications or domain expertise. As described above, few works use CLIP for specific perceptual tasks. However, they either trained all of its parameters, risked the loss of its priors or used a restrictive approach by applying only prompt tuning. In contrast, our approach strikes a balance between preserving CLIP's perceptual prior and allowing task-specific adaptation, achieving state-of-the-art results.

## 3 A Unified Framework for Perceptual Tasks

We propose a *unified framework* that *adapts effectively to diverse perceptual tasks* while maintaining a simple, consistent architecture (see fig. 2). Our approach uses the CLIP vision encoder (Radford et al., 2021), followed by an MLP head for task-specific predictions. To preserve CLIP's pretrained knowledge, we apply LoRA (Hu et al., 2021) to the attention weights, enabling lightweight task-specific adaptation with minimal additional parameters. We use the same architecture and set of hyperparameters across all tasks, reducing the need for extensive task-specific adjustments. In section 5, we evaluate our approach on three distinct perceptual tasks: Visual Emotion Analysis, Memorability Prediction, and Image Quality Assessment.

### A Unified Architecture Across Perceptual Tasks
#### Finetuned Per Task and Dataset

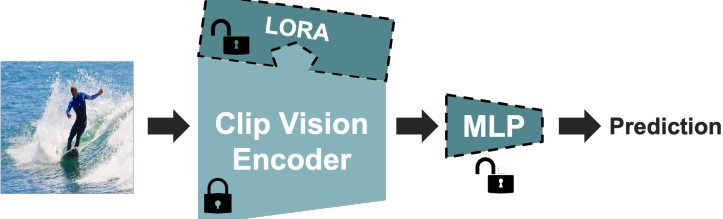

Figure 2: *Unified Framework for Perceptual Tasks: Leveraging the CLIP vision encoder, following an MLP head, our architecture maintains a simple, shared structure across diverse perceptual tasks. With lightweight LoRA adaptation, it fine-tunes efficiently for each task independently, effectively exploiting CLIP's prior perceptual knowledge.*

### 3.1 Architecture and Training

To enable efficient adaptation while preserving CLIP's perceptual knowledge, we fine-tune only the query (q), key (k), and value (v) attention layers of the vision transformer using LoRA, with r=16 (decomposition rank) and $\alpha$=8 (scaling factor). This significantly reduces the number of trainable parameters while allowing task-specific adaptations, resulting in *less than* 3M trainable parameters.

**Model Training** We follow a consistent training strategy across all tasks. The LoRA weights and the MLP are optimized using the AdamW optimizer with a weight decay of 1e-4. We explore 4 different learning rates (5e-5, 1e-4, 5e-4, 1e-3) and implement early stopping with a patience of 12 epochs and maximum of 40 epochs. The final model is selected based on the highest validation score. Before being input into the model, all images are normalized using the CLIP preprocessor. Task specific losses and augmentations are explained in section 4 and Appendix A.1.

### 3.2 Multi Dataset Training

While our method performs well, generalizing from training to testing can still be challenging with small datasets. We propose adapting the same CLIP vision encoder using LoRA across multiple datasets within the same task, allowing the model to leverage larger datasets and improve performance on smaller ones (see section 5).

Our approach uses a single CLIP vision encoder with LoRA weights, along with dataset-specific MLP heads. The training procedure follows a two-stage process. In the first stage, the entire model is trained together, with proportional sampling ensuring that each dataset contributes to a batch in proportion to its size. The model is then optimized by averaging dataset-specific losses. In the second stage, only the MLP heads are fine-tuned towards the target dataset while the CLIP encoder remains frozen. This allows our model to adapt to each dataset with its unique characteristics without altering the shared representations provided by the CLIP encoder (which is shared by all datasets).

## 4 Tasks And Datasets

In this section, we present the three perceptual tasks which were used in our work (fig. 3), along with their dataset details and loss functions (with further details in Appendix A.1).

### 4.1 No-Reference Image Quality Assessment

Image Quality Assessment (IQA) measures how humans perceive image quality, considering factors like sharpness, noise, and distortions, and is important for many visual tasks (e.g. image generation and enhancement). It is formulated as a regression problem, predicting a continuous quality score that aligns with human judgment (see Fig. 3a). We use diverse benchmark datasets with both authentic distortions (e.g., motion blur, overexposure) and synthetic distortions (e.g., manually added noise and color quantization) of varying types and intensities. Each dataset is annotated via crowd-sourced quality ratings, with mean opinion scores (MOS) reflecting human perception.

We used four ***authentic distortion*** benchmark datasets: LIVEC (Ghadiyaram & Bovik, 2015) (1,162 images) and SPAQ (Fang et al., 2020) (11,125 images), both focused on real images captured by mobile devices; KonIQ-10k (Hosu et al., 2020) (10,073 images), which includes diverse real-world scenes; and FLIVE (Ying et al., 2020), comprising 39,810 images from social media and streaming platforms. We further used three ***synthetic distortions*** benchmark datasets, which include distortions such as compression artifacts, noise, and color quantization. Each dataset consists of reference images and corresponding controlled degradations: TID2013 (Ponomarenko et al., 2013) contains 25 reference images and 3,000 distorted images, LIVE (Sheikh et al., 2006) includes 29 reference images and 779 distorted images, and KADID-10k (Lin et al., 2019) provides 81 reference images and 10,125 distorted images.

We follow the same training procedure as Xu et al. (2024a), including the loss function, augmentations, and data splits (see Appendix A.1). Specifically, we use a loss function based on the Pearson Linear Cor-

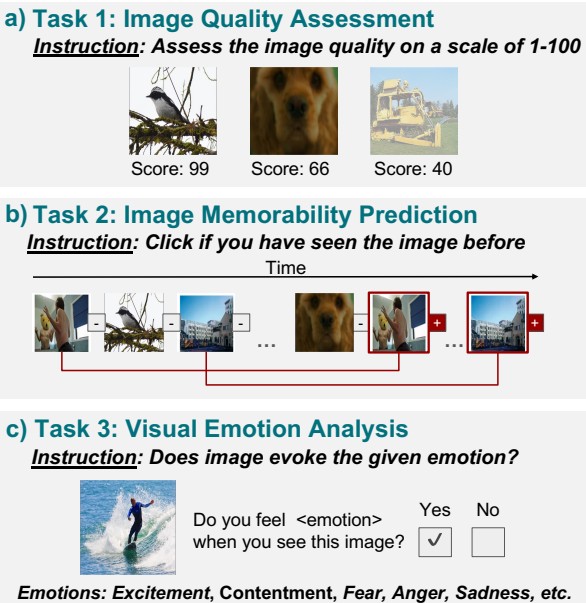

Figure 3: *Visual Perceptual Tasks.*

relation Coefficient (PLCC). Given a batch of m images, with predicted quality scores $\tilde{y} = \tilde{y}_1, \tilde{y}_2, ..., \tilde{y}_m$ and corresponding ground-truth labels $y = y_1, y_2, ..., y_m$, the PLCC-induced loss is defined as: $L_{\text{PLCC}} = \frac{1}{2}(1 - \text{PLCC}(\tilde{y}, y))$ where $\text{PLCC}(\tilde{y}, y)$ represents the Pearson correlation coefficient between the predicted and ground-truth scores. During inference, we average the score predictions from 15 randomly cropped versions of each image. The data is split 10 times, with each split using 80% for training and 20% for testing. From the training set, 10% is further reserved for validation.

## 4.2 Image Memorability Prediction

This task aims to predict how well an image is remembered. It is formulated as a regression problem, where the goal is to estimate a memorability score reflecting the likelihood of the image being remembered (see Fig. 3b).

We train and evaluate our model on two benchmark datasets. The LaMem dataset (Khosla et al., 2015) contains a large-scale collection of ~60,000 images from diverse sources, including scenes, objects, and art. The Things memorability dataset (Kramer et al., 2023) comprises of 26,107 object images covering 1,854 distinct concepts. The LaMem dataset is split into five parts: 75% for training, 5% for validation, and 20% for testing. For the Things dataset, we use the entire dataset for generalization assessment. The loss used for training is the mean squared error (MSE) loss on the memorability score.

## 4.3 Visual Emotion Analysis

Visual Emotion Analysis (VEA) is a classification task focused on predicting the emotional response that an image is likely to evoke in viewers (see Fig. 3c).

We experiment with two prominent benchmark datasets, both supporting binary emotion classification (positive vs. negative) and multi-class classification of more nuanced emotions, carefully labeled by human annotators. EmoSet (Yang et al., 2023), the largest existing visual emotion dataset, contains 118,102 images from social networks and artistic sources, labeled into 8 balanced emotions: Awe, Excitement, Amusement, and Contentment (positive) / Anger, Sadness, Disgust, and Fear (negative). EmotionROI (Peng et al., 2016) includes 1,980 Flickr annotated images, categorized into 6 emotions: Joy, Surprise (positive) / Anger, Disgust, Fear, and Sadness (negative). Following Yang et al. (2023), EmoSet is split into 80% training, 15% test, and 5% validation, while EmotionROI is randomly divided into 5 splits with 75% training, 20% test, and 5% validation. Our model is trained using the cross-entropy loss.

Table 1: **Model Performance on IQA Benchmarks.** *Models are evaluated on seven common Image Quality Assessment benchmarks, reporting the median SRCC (Spearman's Rank Correlation) and PLCC (Pearson Linear Correlation) across 10 splits, along with the number of trainable parameters. The table shows that our model, PerceptCLIP, outperforms all leading IQA-dedicated methods, achieving the best performance, on six out of seven datasets while using significantly fewer trainable parameters.*

| Method | LIVEC | | KonIQ-10k | | SPAQ | | FLIVE | | LIVE | | TID2013 | | KADID-10k | |
|---|---|---|---|---|---|---|---|---|---|---|---|---|---|---|
| | SRCC | PLCC | SRCC | PLCC | SRCC | PLCC | SRCC | PLCC | SRCC | PLCC | SRCC | PLCC | SRCC | PLCC |
| HyperIQA (27M) | 0.852 | 0.882 | 0.906 | 0.917 | 0.911 | 0.915 | 0.544 | 0.602 | 0.962 | 0.966 | 0.840 | 0.858 | 0.852 | 0.845 |
| MUSIQ (27M) | 0.702 | 0.746 | 0.916 | 0.928 | 0.918 | 0.921 | 0.566 | 0.661 | 0.940 | 0.911 | 0.773 | 0.815 | 0.875 | 0.872 |
| TREs (152M) | 0.846 | 0.877 | 0.915 | 0.928 | - | - | 0.544 | 0.625 | 0.969 | 0.968 | 0.863 | 0.883 | 0.859 | 0.858 |
| DEIQT (24M) | 0.875 | 0.894 | 0.921 | 0.934 | 0.919 | 0.923 | 0.571 | 0.663 | **0.980** | **0.982** | **0.892** | **0.908** | 0.889 | 0.887 |
| LIQE (151M) | **0.904** | 0.910 | 0.919 | 0.908 | - | - | - | - | 0.970 | 0.951 | - | - | 0.930 | 0.931 |
| Re-IQA (24M) | 0.840 | 0.854 | 0.914 | 0.923 | 0.918 | 0.925 | **0.645** | **0.733** | 0.970 | 0.971 | 0.804 | 0.861 | 0.872 | 0.885 |
| QPT-ResNet50 (24M) | 0.895 | **0.914** | 0.927 | 0.941 | 0.925 | 0.928 | 0.575 | 0.675 | - | - | - | - | - | - |
| LoDa (9M) | 0.876 | 0.899 | **0.932** | **0.944** | **0.925** | **0.928** | 0.578 | 0.679 | 0.975 | 0.979 | 0.869 | 0.901 | **0.931** | **0.936** |
| **PerceptCLIP (Ours) (3M)** | **0.908** | **0.923** | **0.945** | **0.954** | **0.930** | **0.933** | 0.590 | 0.689 | **0.980** | **0.982** | 0.887 | **0.908** | **0.952** | **0.956** |

# 5 Experimental Results

In this section, we present the results of our proposed approach for training models on three different perceptual tasks. Our trained models achieved state-of-the-art (SOTA) results with significant improvements across all three tasks. We first present the results for each task, comparing them against the best task-specific methods. We then demonstrate the effectiveness of training a model with multiple datasets for a given task, leading to substantial performance gains on smaller datasets.

## 5.1 Image Quality Assessment

We train and evaluate our model on 7 prominent datasets, covering both authentic and synthetic distortions, using the standard 10 splits provided by Xu et al. (2024a). Our approach is evaluated using the median Spearman's Rank Correlation Coefficient (SRCC) and median Pearson's Linear Correlation Coefficient (PLCC) with a 4-parameter logistic regression correction, as in Xu et al. (2024a) (see Appendix A.1).

Table 1 summarizes our model's performance in predicting image quality scores compared to leading IQA-dedicated methods (Su et al., 2020; Ke et al., 2021; Kramer et al., 2023; Golestaneh et al., 2022; Qin et al., 2023; Zhang et al., 2023; Saha et al., 2023; Zhao et al., 2023; Xu et al., 2024a), along with the number of trainable parameters for each model. Our model achieves the best results on 5 out of 7 datasets in both SRCC and PLCC. Our results are even more impressive when considering the number of trainable parameters. Our model consist approximately 3M trainable parameters, compared to 9M to 152M in other models, highlighting the strength of CLIP's perceptual prior.

Furthermore, as shown in Table 2, our method not only sets new state-of-the-art results but also demonstrates strong generalization capabilities *across different datasets* (i.e., training on one dataset, but testing on another). This highlights the robustness of our model and its ability to learn meaningful perceptual features, rather than just overfitting dataset-specific patterns. Notably, the improvement in generalization is significantly larger than the improvement in within-dataset evaluation. For instance, we achieve a 1.4% relative SRCC improvement compared to LoDA when training and testing on KonIQ, while in a generalization setting, when training on FLIVE and testing on KonIQ, the improvement increases to 6.4% relative SRCC. Our lightweight adaptation ensures that we retain CLIP's strong human-like perceptual priors when fine-tuning for specific tasks.

## 5.2 Image Memorability Prediction

We train and evaluate our model on the LaMem dataset using the standard 5-fold validation strategy. To assess generalization, we evaluate our model on the THINGS memorability dataset, as done in Needell & Bainbridge (2022); Hagen & Espeseth (2023). Performance is reported using the mean SRCC and averaged mean squared error (MSE) across 5 folds. Table 3 presents our model's performance on LaMem compared to

Table 2: ***Cross-Dataset Generalization in IQA.*** *SRCC generalization results for training on one dataset and testing on another. PerceptCLIP significantly outperforms all previous models.*

| Train dataset | FLIVE | | LIVEC | KonIQ |
|---|---|---|---|---|
| **Test dataset** | **KonIQ** | **LIVEC** | **KonIQ** | **LIVEC** |
| HyperIQA | 0.758 | 0.735 | 0.772 | 0.785 |
| TReS | 0.713 | 0.740 | 0.733 | 0.786 |
| DEIQT | 0.733 | 0.781 | 0.744 | 0.794 |
| QPT-ResNet50 | - | - | 0.749 | 0.821 |
| LoDa | 0.763 | 0.805 | 0.745 | 0.811 |
| **PerceptCLIP (Ours)** | **0.812** | **0.825** | **0.794** | **0.875** |

leading memorability-dedicated methods (Khosla et al., 2015; Fajtl et al., 2018; Leonardi et al., 2019; Hagen & Espeseth, 2023; Squalli-Houssaini et al., 2018). Our model significantly surpasses all others, achieving new state-of-the-art results on both metrics. Specifically, it improves SRCC by 2.9% relative to the second-best model (Squalli-Houssaini et al., 2018) and reduces MSE by 9.2% relative to the previous best-performing model (Hagen & Espeseth, 2023). Furthermore, our model demonstrates significantly better generalization capabilities. When trained on LaMem and evaluated on THINGS, it achieves a 12% relative improvement over the best previously reported result (Hagen & Espeseth, 2023), as shown in the Table A.1. Results on another small image memorability dataset, SUN (Isola et al., 2011b), can be seen in Table A.2. Altogether, this demonstrates the power of CLIP as a prior for the image memorability task.

## 5.3 Visual Emotion Prediction

We train and evaluate on EmoSet and EmotionROI datasets, using the standard splits as described in Section 4.3, and report emotion classification accuracy for 2, 6, and 8 emotion categories, depending on the dataset. Table 4 compares our model's performance with various existing methods (Rao et al., 2019; Xu et al., 2022; Yang et al., 2021b; Deng et al., 2022; 2024; Xu et al., 2024b) designed to classify the emotions evoked by images. The results demonstrate that our model achieves state-of-the-art performance on both benchmarks, excelling in both binary emotion classification (positive vs. negative) and multi-class emotion classification (8 categories for EmoSet, and 6 for EmotionROI).

Notably, the improvement in multi-class emotion classification is more significant, as binary classification already exhibits high performance across models. Our model achieves an approximately 4% relative accuracy improvement for both EmoSet and EmotionROI in multi-class emotion classification. These results further strengthen the effectiveness of our approach. Additional comparisons and metrics can be found in Appendix A.2, in Table A.3.

Table 3: ***Model Performance on Image Memorability.*** *Average SRCC and MSE across 5 splits of the LAMEM dataset are reported, showing that our model, PerceptCLIP, surpasses all previous methods.*

| Method | LaMem | |
|---|---|---|
| | SRCC↑ | MSE↓ |
| MemNet | 0.640 | Unknown |
| AMNet | 0.677 | 0.0082 |
| Leonardi et al | 0.687 | 0.0079 |
| ViTMem | 0.711 | 0.0076 |
| Squalli-Houssaini et al | 0.720 | 0.0092 |
| **PerceptCLIP (Ours)** | **0.741** | **0.0069** |

Table 4: ***Model Performance on Emotion Classification.*** *Accuracy results for binary and multi-class emotion classification on the EmotionROI (mean over five splits) and EmoSet datasets. Our model achieves state-of-the-art performance for both datasets.*

| Method | EmoSet | | EmotionROI | |
|---|---|---|---|---|
| | 8 | 2 | 6 | 2 |
| MRCNN | 0.7539 | 0.9228 | - | - |
| MDAN | 0.7575 | 0.9371 | 0.6166 | - |
| StimuliVEA | 0.7840 | 0.9458 | 0.6162 | - |
| PT-DPC | 0.7713 | 0.9246 | 0.6970 | 0.8855 |
| SimEmotion | 0.7906 | 0.9428 | 0.7054 | 0.9040 |
| MVP | 0.8192 | 0.9644 | 0.7189 | 0.9255 |
| **PerceptCLIP (Ours)** | **0.8528** | **0.9780** | **0.7485** | **0.9288** |

Table 5: ***Multi-Dataset Training Improves Performance.*** *The table shows the benefits of training on multiple datasets within the same task, demonstrating significantly improved performance on smaller datasets. Results for all datasets (shown in the Appendix) show that these multi-dataset models achieve state-of-the-art results across all benchmarks.*

| Method | IQA | | | | VEA | Memorability Prediction | |
|---|---|---|---|---|---|---|---|
| | LIVEC | | TID2013 | | EmoROI-6 | THINGS | |
| | SRCC↑ | PLCC↑ | SRCC↑ | PLCC↑ | ACC↑ | SRCC↑ | MSE↓ |
| Previous SOTA | 0.904 | 0.910 | 0.892 | 0.908 | 0.7189 | - | - |
| PerceptCLIP (Ours) | 0.908 | 0.923 | 0.887 | 0.908 | 0.7485 | 0.452 | 0.0058 |
| PerceptCLIP Multi-Dataset (Ours) | **0.922** | **0.933** | **0.900** | **0.915** | **0.7591** | **0.454** | **0.0054** |

### 5.4 Multi-Dataset Training

We present the results of our multi-dataset training, where a single model is trained on multiple datasets of the same task. In this setup, the LoRA finetuning of CLIP's attention layers is shared across datasets, while the final added MLP heads are dataset-specific.

Training on multiple datasets provides additional data, significantly improving performance on smaller datasets (see Table 5) while maintaining comparable results on larger datasets (see Tables A.4 to A.6). We present the results of four models, each trained on a specific set of datasets: (1) authentic IQA datasets (LIVEC, KonIQ-10k, SPAQ), (2) synthetic IQA datasets (KADID-10K, LIVE, TID-2013), (3) memorability datasets (THINGS, LaMem), and (4) emotion datasets (EmotionROI, EmoSet). For example, in the IQA task we achieve a 1.5% relative improvement on the small LIVEC dataset (which contains only 1,126 examples), when training together with the larger KonIQ and SPAQ datasets, compared to training on LIVEC alone. These results emphasize the power of using CLIP as a prior with minimal adaptation, as it allows us to leverage CLIP's perceptual knowledge, which is likely well-aligned with the shared properties across different datasets. When comparing our multi-dataset training against previous models on these datasets, the improvement is even more significant (e.g., a relative improvement of 5.6% over the previous SOTA on EmoROI-6). Our multi-dataset models achieve improved SOTA results across all datasets and tasks.

## 6 Interpretability and Visualizations

We visualize our model's attention heads to gain deeper insight into its decision-making process. Specifically, we focus on analyzing attention across all layers to identify the heads most critical for prediction. To achieve this, we employ an automated attention masking approach to systematically determine the importance of individual heads. First, an image is passed through the model to obtain a baseline prediction. Then, the same image is processed again, but this time the CLS token's attention for the target head is replaced with a uniform, equal-weighted map. By comparing the resulting prediction with the baseline (i.e., before and after masking), we can determine whether the head's attention is significant for prediction. A significant drop in performance indicates that the head is important. Repeating this process over many images and averaging the prediction differences allows us to quantify the importance of each head.

For the most influential (automatically-detected) heads, we select images that exhibit the largest prediction differences. A few such images are presented in fig. 4, along with differences between attention heatmaps of PerceptCLIP and the original CLIP. This comparison highlights the shift in attention, revealing how our fine-tuned model reallocates focus to different image regions, revealing the features it considers relevant for perceptual property prediction.

Figure 4 displays visual examples for both emotion and memorability predictions. The attention maps show that the model attends to semantically relevant regions for each emotion. For instance, in the image classified as fear-inducing, the model focuses on blood, a strong visual cue for fear. For anger, it attends to fire, while for excitement, it highlights the girl with an open mouth and a 'like' hand gesture (see more examples in Figures A.1 and A.2). For memorability, the model mainly focuses on the most distinctive and recognizable

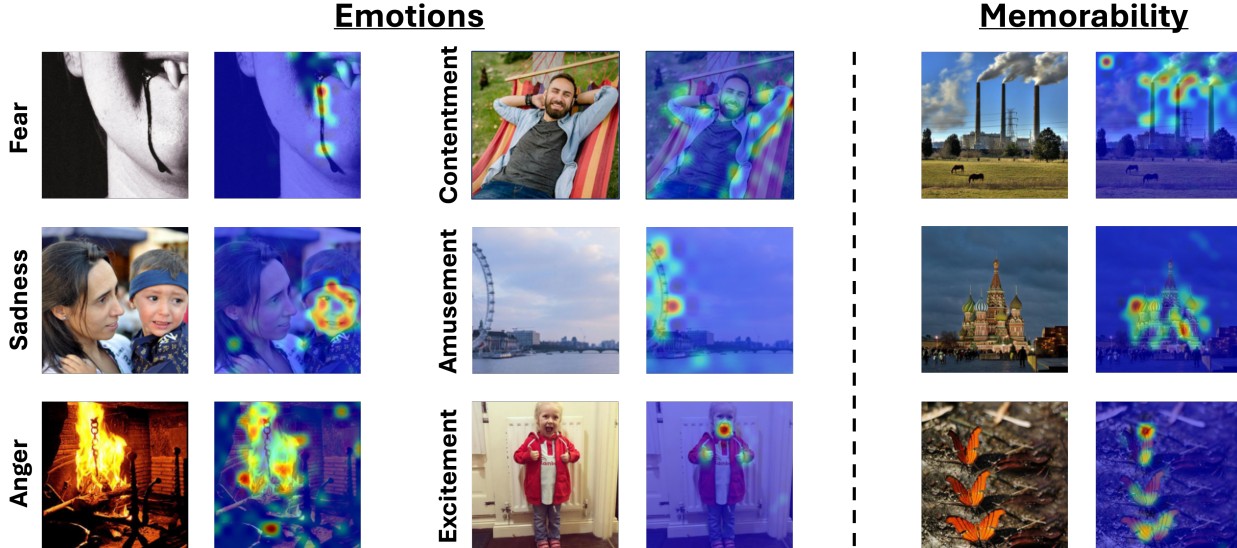

Figure 4: ***Attention Shift Toward Perceptual Cues.*** *We present images along with the differences in their attention maps between our PerceptCLIP model and the pretrained CLIP vision encoder (displaying results from critical attention heads that most influence the perceptual predictions). This highlights the shift in attention, revealing how our model reallocates focus to perceptually meaningful regions.*

objects in the image. For example, in an image containing multiple elements, the model attends to chimneys with smoke, which may contribute significantly to memorability. Altogether, these demonstrate that our fine-tuned model successfully learns to focus on perceptually meaningful image features.

## 7 Ablation Study

In this section, we conduct an ablation study to analyze the impact of different design choices in our approach.

**Training Strategy.** We aim to investigate how to effectively leverage CLIP's perceptual prior while adapting it to different tasks. We compare three different training strategies: (1) full fine-tuning of CLIP's vision encoder, (2) freezing CLIP and training only the MLP head, and (3) our chosen approach – fine-tuning CLIP by using LoRA adaptation (of the attention heads). As shown in Table 6, full fine-tuning significantly degrades the performance. This is likely due to catastrophic forgetting, where the model loses its strong pre-trained prior and fails to generalize effectively. While training only the MLP (with frozen CLIP weights) yields better results than fully fine-tuning CLIP, it still falls short compared to using LoRA. Using LoRA adaptation provides the best balance, allowing the model to fine-tune efficiently while preserving CLIP's strong perceptual prior. Ablation on our two-step multi-dataset training can be found in the Appendix A.3.

Table 6: ***Impact of Training Strategies.*** *We compare three training strategies for the CLIP visual encoder—full fine-tuning, freezing, and LoRA adaptation. The LoRA-based approach (which is used in our framework) outperforms both full fine-tuning and freezing, striking the best balance between adaptation and preserving CLIP's perceptual prior.*

| Training Procedure | IQA KonIQ-10k | | VEA EmoSet-8 | Memorability LaMem | |
|---|---|---|---|---|---|
| | SRCC↑ | PLCC↑ | ACC↑ | SRCC↑ | MSE↓ |
| **Full Fine-tuning CLIP** | 0.888 | 0.904 | 0.715 | 0.618 | 0.0466 |
| **Frozen CLIP** | 0.894 | 0.913 | 0.845 | 0.691 | 0.0081 |
| **LoRA for CLIP** | **0.945** | **0.954** | **0.853** | **0.741** | **0.0069** |

**MLP vs. Linear Layer** We explored the impact of different head architectures by comparing a simple linear layer with the MLP head used in our main model. As shown in Table 7, the choice of head has a minor effect on performance in the IQA and VEA tasks, with both architectures performing competitively. However, the MLP head demonstrates better performance on the memorability task. In all tasks, both heads consistently achieve SOTA results using our approach.

Table 7: ***Effect of Different Heads on Performance.*** *We compared the performance of a simple linear layer and the MLP head used in our main model. The choice of head has a minor effect on performance in the IQA and VEA tasks. However, the MLP head shows better performance on the memorability task.*

| Method | IQA
KonIQ-10k | | VEA
EmoSet-8 | Memorability
LaMem | |
|---|---|---|---|---|---|
| | SRCC↑ | PLCC↑ | ACC↑ | SRCC↑ | MSE↓ |
| **Linear Layer** | 0.943 | 0.953 | 0.8526 | 0.734 | 0.0077 |
| **MLP (Ours)** | **0.945** | **0.954** | **0.8528** | **0.741** | **0.0069** |

**Backbone Choice.** To assess the role of different vision backbones as perceptual priors, we experimented with 3 different pre-trained backbone models (MAE, DINOv2, and CLIP, all using ViT-L/14), while keeping the rest of the architecture and LoRA configuration the same. As shown in Table A.8, while our approach performs well also with DINOv2, CLIP achieves the best results across all three tasks. This provides further evidence that CLIP effectively captures perceptual properties, making it a strong prior for perceptual tasks.

**Effect of ViT Size.** To assess the impact of vision transformer size on performance, we compared three CLIP vision encoder variants: ViT-B/16, ViT-B/32, and ViT-L/14. The results, summarized in Table A.7, indicate that ViT-B/16 outperforms ViT-B/32, suggesting that a finer patch resolution contributes to better perceptual understanding. Additionally, ViT-L/14 achieves the best results across all tasks, reinforcing our choice to use it as our backbone.

## 8 Conclusion

We propose a simple unified framework for visual perceptual tasks, which leverages the implicit yet rich *perceptual knowledge* of CLIP. The human-annotated image captions in CLIP's training data contain also human sentiments and emotions, which is what probably makes CLIP a particularly strong prior for *human judgment*. We apply minimal task-specific adaptations, aiming to balance preserving CLIP's strong perceptual prior, while allowing necessary task-specific flexibility. Remarkably, this lightweight yet effective framework produces models that achieve state-of-the-art performance on 3 important perceptual tasks, as well as demonstrates impressive cross-dataset generalization capabilities. This suggests that complex, task-specific designs and knowledge are not necessarily required for modeling perceptual tasks.

**Acknowledgment:** This project was funded by the European Union (ERC, MindReading, 101142115).

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

# A  Appendix

## A.1  Further Technical Details

**Data Augmentations.**  In our training we used a common data augmentation techniques which are detailed below. **Image Quality Assessment:** Following  Xu et al. (2024a), during training, each image is augmented into 3 to 10 replicas, depending on the dataset. Each replica undergoes independent random horizontal and vertical flips (p = 0.5) and random cropping to 224×224. **Predicting Image Memorability:** Before feeding the images into the model, we resize the shortest side of each image to 224 pixels while maintaining the aspect ratio. A center crop of size 224×224 is then applied. **Visual Emotion Analysis:** We apply data augmentation inspired by  Yang et al. (2023), including random resized cropping to 224×224 and horizontal flipping (p = 0.5). During inference, images are resized so that their shorter side is 224 pixels while preserving the aspect ratio, followed by a center crop.

**Training Runtime**  Training on an NVIDIA L40S GPU takes up to a day for the largest dataset (e.g., EmoSet with approximately 100,000 images) and only a few hours for smaller datasets when run for 40 epochs. In practice, early stopping significantly reduces the overall training time.

**4-parameter logistic regression correction.**  For the Image quality assessment task, we report Pearson's Linear Correlation Coefficient (PLCC) with a 4-parameter logistic regression correction, as proposed by Antkowiak et al. (2000) and following  Xu et al. (2024a). The 4-parameter logistic function is defined as:

$$y' = \beta_2 + \frac{\beta_1 - \beta_2}{1 + e^{-\left(\frac{x - \beta_3}{|\beta_4|}\right)}} \tag{1}$$

where $y'$ is the transformed prediction, $x$ is the original model output, and $\beta_1$, $\beta_2$, $\beta_3$, and $\beta_4$ are the four parameters. This function is fitted to the model predictions using non-linear regression, optimizing its parameters to best match the subjective scores. Once the predictions are adjusted using this transformation, PLCC is computed to measure the correlation between the corrected predictions and human ratings.

## A.2  Additional Results

### A.2.1  Image Memorability Additional Results

Table A.1 illustrates the generalization capability of our model compared to two other models on the memorability task. ResMemNeedell & Bainbridge (2022) and ViTMemHagen & Espeseth (2023) were trained using a combined dataset of LaMem (58,741 images) Khosla et al. (2015) and MemCat (10,000 images) Goetschalckx & Wagemans (2019), while our model was trained exclusively on the LaMem dataset. All models were evaluated on the Things memorability dataset. Our model outperforms the others, demonstrating superior generalization capabilities.

Table A.1: *Generalization Results on Things Dataset.*

| Method | THINGS SRCC |
|---|---|
| resMem | 0.22 |
| ViTMem | 0.30 |
| **PerceptCLIP (Ours)** | **0.34** |

Table A.2: **Model Performance on SUN Image Memorability dataset.** Average SRCC and MSE across 25 splits of the SUN dataset are reported, showing that our model, PerceptCLIP, surpasses all previous methods.

| Method | SUN | |
|---|---|---|
| | SRCC↑ | MSE↓ |
| MemNet | 0.610 | unknown |
| AMNet | 0.649 | **0.011** |
| **PerceptCLIP (Ours)** | **0.708** | 0.014 |

### A.2.2 Visual Emotion Analysis - Additional Metrics

In addition to the results presented in the paper, we provide additional evaluation metrics in table A.3, comparing our model with two leading existing models. These metrics include: ***Mean Average Precision (mAP):*** Measures the area under the precision-recall curve, assessing the model's ability to rank positive examples higher than negatives across different thresholds. ***Macro F1 Score (m-F1):*** Computes the F1 score for each class independently and averages them, treating all classes equally regardless of dataset imbalance. ***Weighted F1 Score (w-F1):*** The harmonic mean of precision and recall, weighted by class support, ensuring that both high precision and recall contribute fairly across imbalanced datasets.

By incorporating these diverse metrics, we ensure a more comprehensive assessment of model performance beyond simple accuracy. Our model consistently outperforms the others across all metrics, demonstrating its robustness and effectiveness in handling complex classification tasks.

Table A.3: ***Additional Metrics for the Emotion Recognition Task:*** *We report Accuracy, mean Average Precision (mAP), macro-F1 (m-F1), and weighted-F1 (w-F1) on the EmoSet and EmotionROI datasets. Our approach outperforms existing methods, achieving the best results across all metrics.*

| Method | EmoSet-8 | | | | EmotionROI-6 | | | |
|---|---|---|---|---|---|---|---|---|
| | Accuracy | mAP | m-F1 | w-F1 | Accuracy | mAP | m-F1 | w-F1 |
| SimEmotion | 0.7906 | 0.7387 | 0.7514 | 0.7894 | 0.7054 | 0.7054 | 0.7082 | 0.7027 |
| MVP | 0.8192 | 0.8003 | 0.8059 | 0.8047 | 0.7189 | 0.7116 | 0.7194 | 0.7148 |
| **Ours** | **0.8528** | **0.9336** | **0.8592** | **0.8526** | **0.7485** | **0.8081** | **0.7432** | **0.7445** |

### A.2.3 Multi-Dataset

We conducted joint training on multiple datasets for the same task, where the CLIP LoRA adaptation is shared, and each dataset has its own MLP. We present the results of these models on all datasets in tables A.4 to A.6. These results show a significant improvement on smaller datasets, leveraging multi-dataset training with our approach, while achieving comparable results on larger datasets. Overall, our models trained on multiple datasets achieve state-of-the-art results across all tasks and datasets.

Table A.4: ***Multi-Dataset vs. Single-Dataset Results on IQA:*** *We compare models trained on individual datasets with those trained on multiple datasets. Multi-dataset models were trained on either authentic distortions (LIVEC, KonIQ-10k, SPAQ) or synthetic distortions (LIVE, TID2013, KADID-10k). Multi-Dataset models improves performance on smaller datasets while maintaining comparable results on larger ones.*

| Method | LIVEC | | KonIQ-10k | | SPAQ | | LIVE | | TID2013 | | KADID-10k | |
|---|---|---|---|---|---|---|---|---|---|---|---|---|
| | SRCC | PLCC | SRCC | PLCC | SRCC | PLCC | SRCC | PLCC | SRCC | PLCC | SRCC | PLCC |
| PerceptCLIP | 0.908 | 0.923 | **0.945** | **0.954** | **0.930** | 0.933 | **0.980** | 0.982 | 0.887 | 0.908 | 0.952 | **0.956** |
| PerceptCLIP Multi-Dataset | **0.922** | **0.933** | 0.944 | 0.953 | **0.930** | **0.934** | **0.980** | **0.983** | **0.900** | **0.915** | **0.953** | **0.956** |

Table A.5: **Multi-Dataset vs. Single-Dataset Training for Memorability Prediction:** *The multi-dataset approach improves performance on THINGS while maintaining comparable results on LaMem.*

| Method | LaMem | | THINGS | |
|---|---|---|---|---|
| | SRCC | MSE | SRCC | MSE |
| PerceptCLIP | **0.741** | **0.0069** | 0.452 | 0.0058 |
| PerceptCLIP Multi-Dataset | 0.740 | **0.0069** | **0.454** | **0.0054** |

Table A.6: **Multi-Dataset vs. Single-Dataset Training for Emotion Recognition:** *Training EmotionROI-6 together with EmoSet-8 improves performance on EmotionROI-6 while maintaining comparable results to single-dataset training on EmoSet-8.*

| Method | EmoSet-8 | EmotionROI-6 |
|---|---|---|
| PerceptCLIP | **0.8528** | 0.7485 |
| PerceptCLIP Multi-Dataset | 0.8511 | **0.7591** |

## A.3 Ablation Study

In this section, we present additional details from our ablation studies, including comprehensive tables, as well as further ablations of the multi-dataset training procedure.

**Different CLIP ViT Sizes.** Table A.7 Shows the results of using different ViT sizes for the CLIP vision encoder backbone in our architecture.

Table A.7: **Performance of Different CLIP Variants:** *We trained our model using different sizes of ViT as the CLIP vision encoder. The results show that ViT-B/16 outperforms ViT-B/32, suggesting that a finer patch resolution improves perceptual understanding. ViT-L/14 achieves the best overall performance, reinforcing its use as our backbone model.*

| Method | IQA KonIQ-10k | | VEA EmoSet-8 | Memorability LaMem | |
|---|---|---|---|---|---|
| | SRCC↑ | PLCC↑ | ACC↑ | SRCC↑ | MSE↓ |
| **ViT-B/32** | 0.932 | 0.944 | 0.8303 | 0.724 | 0.0073 |
| **ViT-B/16** | 0.938 | 0.949 | 0.8422 | 0.731 | 0.0071 |
| **ViT-L/14** | **0.945** | **0.954** | **0.8528** | **0.741** | **0.0069** |

**Different Vision Backbones.** Table A.8 shows the effect of different vision backbones on performance (including TULIP model)

Table A.8: **Effect of Vision Backbones on Performance:** *We trained our model with different backbones, applying LoRA to the q, k, and v layers. CLIP outperforms MAE and DINOv2 across all tasks, demonstrating its stronger perceptual prior.*

| Backbone | IQA KonIQ-10k | | VEA EmoSet-8 | Memorability LaMem | |
|---|---|---|---|---|---|
| | SRCC↑ | PLCC↑ | ACC↑ | SRCC↑ | MSE↓ |
| **MAE** | 0.924 | 0.935 | 0.7681 | 0.699 | 0.0078 |
| **DinoV2** | 0.937 | 0.950 | 0.8252 | 0.729 | 0.0071 |
| **TULIP** | 0.942 | 0.950 | 0.841 | 0.682 | 0.0076 |
| **CLIP** | **0.945** | **0.954** | **0.8528** | **0.741** | **0.0069** |

**Multi-Dataset Training** In our multi-dataset setting, we use two-step training procedure, where we first train the shared CLIP LoRA weights and the MLP for each dataset together with all datasets. In the second step, each dataset's MLP is trained separately while the CLIP LoRA weights are fixed and not optimized. We have conducted an ablation study in table A.9 to demonstrate that the performance improvement is not solely due to the two-phase training procedure. We compare three approaches: (1) training a separate model for each dataset using our single-dataset approach, (2) training a separate model for each dataset using a two-phase procedure (first training the entire model, then fine-tuning only the MLP while keeping

CLIP frozen), and (3) training a single model on multiple datasets using our multi-dataset approach with the two-phase training. The multi-dataset approach leads to better performance, while the two-phase training with a single dataset yields comparable results to the single-dataset approach.

Table A.9: *Ablation Study for Multi-Dataset Training.*

| Method | IQA | | | | VEA | Memorability prediction | |
|---|---|---|---|---|---|---|---|
| | LIVEC | | TID2013 | | EmoROI-6 | THINGS | |
| | SRCC↑ | PLCC↑ | SRCC↑ | PLCC↑ | ACC↑ | SRCC↑ | MSE↓ |
| **PerceptCLIP** Single-Dataset | 0.908 | 0.923 | 0.887 | 0.908 | 0.7485 | 0.452 | 0.0058 |
| **PerceptCLIP** 2-Phase Training Single-Dataset | 0.907 | 0.923 | 0.889 | 0.909 | 0.7500 | 0.451 | 0.0055 |
| **PerceptCLIP** 2-Phase Training Multi-Dataset | **0.922** | **0.933** | **0.900** | **0.915** | **0.7591** | **0.454** | **0.0054** |

**LoRA Parameters**   We have systematically experimented with different values of $r$ and $\alpha$ on our main model using the KonIQ-10k dataset from the IQA task (see tables A.10 and A.11). As shown in those tables, these parameters have minimal impact on model performance.

Table A.10: **SRCC Performance Across LoRA Configurations:** Spearman Rank Correlation Coefficient (SRCC) values for different combinations of LoRA rank ($r$) and scaling factor ($\alpha$) on the **kon10k dataset** within the IQA task.

| r$\alpha$ | 4 | 8 | 16 |
|---|---|---|---|
| **8** | 0.945 | 0.944 | 0.944 |
| **16** | 0.944 | 0.945 | 0.945 |
| **32** | 0.945 | 0.944 | 0.944 |

Table A.11: **PLCC Performance Across LoRA Configurations:** Pearson Linear Correlation Coefficient (PLCC) values for different combinations of LoRA rank ($r$) and scaling factor ($\alpha$) on the **kon10k dataset** within the IQA task.

| r$\alpha$ | 4 | 8 | 16 |
|---|---|---|---|
| **8** | 0.954 | 0.954 | 0.954 |
| **16** | 0.954 | 0.954 | 0.954 |
| **32** | 0.954 | 0.953 | 0.953 |

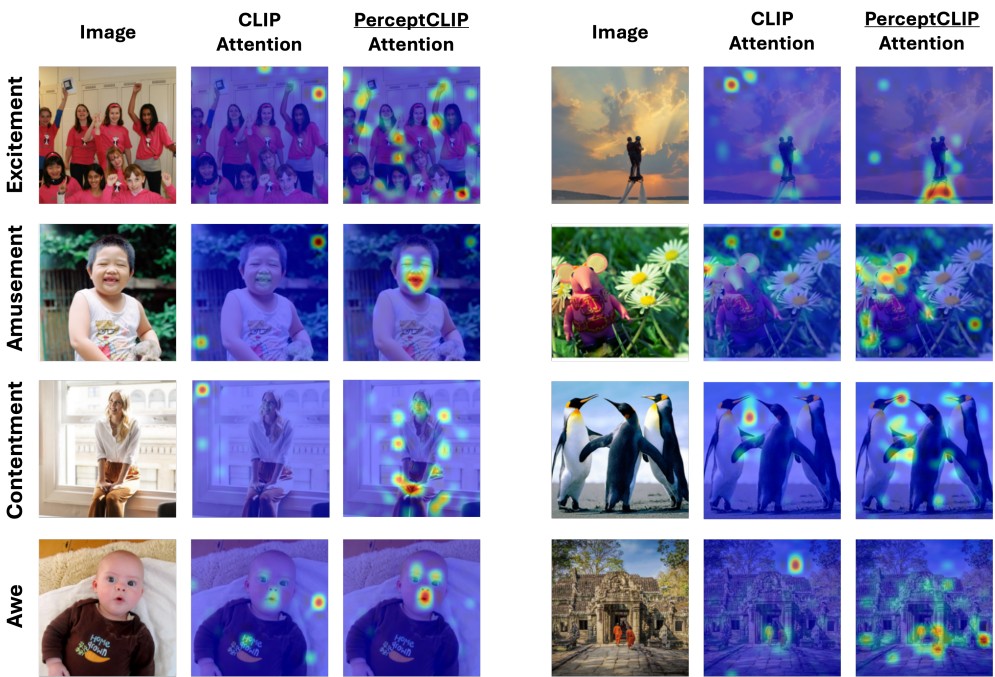

Figure A.1: **Attention Mask Visualization - Positive Emotions.** *We present images alongside their corresponding attention maps from the pretrained CLIP vision encoder and our PerceptCLIP model (displaying results from critical attention heads that most influence the perceptual predictions). This highlights the shift in attention, revealing how our model reallocates focus to perceptually meaningful regions.*

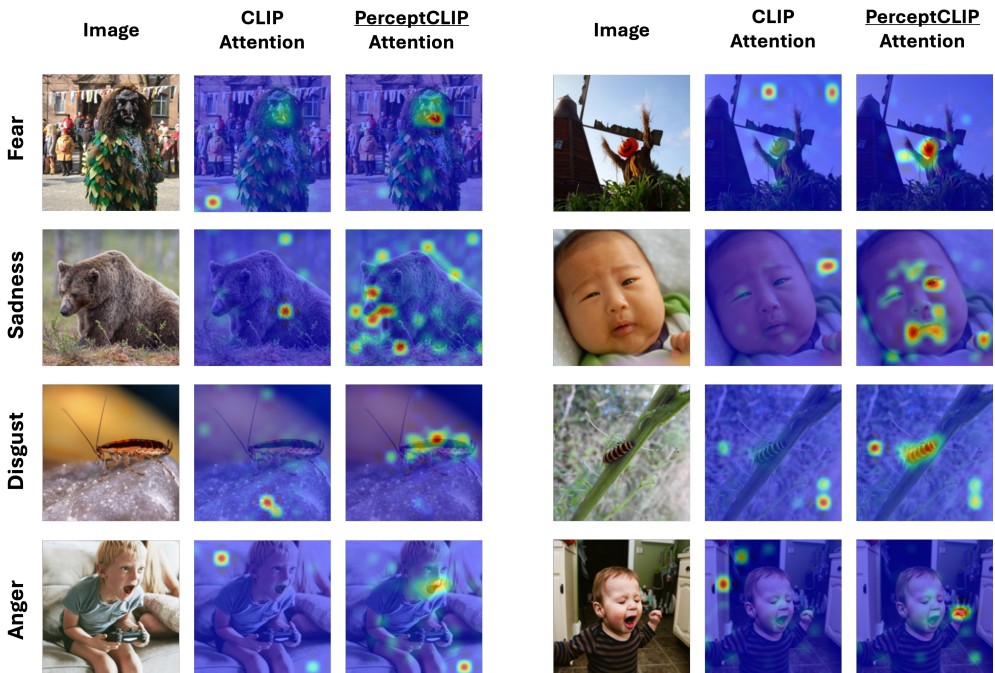

Figure A.2: **Attention Mask Visualization - Negative Emotions.** *We present images alongside their corresponding attention maps from the pretrained CLIP vision encoder and our PerceptCLIP model (displaying results from critical attention heads that most influence the perceptual predictions). This highlights the shift in attention, revealing how our model reallocates focus to perceptually meaningful regions.*

### A.4 Error Analysis and Model Limitations

**Visual Emotion Analysis.** Figure A.3 presents the confusion matrix for PerceptCLIP's classification performance on EmoSet. Most misclassifications occurred between semantically similar emotions (e.g., amusement vs. contentment, fear vs. disgust)—confusions that are also common among human annotators (e.g., do we fear bugs or feel disgusted by them?).

A notable cross-valence confusion was observed between sadness (a negative emotion) and contentment (a positive one). As shown in Figure A.4, many of these images feature animals, suggesting a potential model bias associating animals with positive emotions. Another frequent confusion occurred between anger and excitement, often in images showing people with open mouths—such as during protests or singing (see Figure A.5). These findings highlight the subjectivity of emotion perception and point to a possible model bias toward interpreting open-mouthed expressions as excitement.

**Memorability Prediction.** FigureA.6 presents a histogram comparing the ground-truth and predicted memorability scores on the LaMEM test set (split 1). We observed that PerceptCLIP's predictions are more concentrated around the mean. To further explore this behavior, we visualized the images with the largest prediction errors at both extremes of the memorability spectrum, as shown in fig. A.7. However, these examples did not reveal any consistent or interpretable visual patterns, making it difficult to pinpoint a specific failure mode.

**Image Quality Assessment.** Figure A.8 shows examples of images with the highest prediction errors in the IQA task for the KonIQ10k dataset. We applied z-score normalization to both the predicted and ground-truth scores before identifying these cases. The failure cases exhibited only subtle patterns, potentially involving complex images that require a high-level understanding.

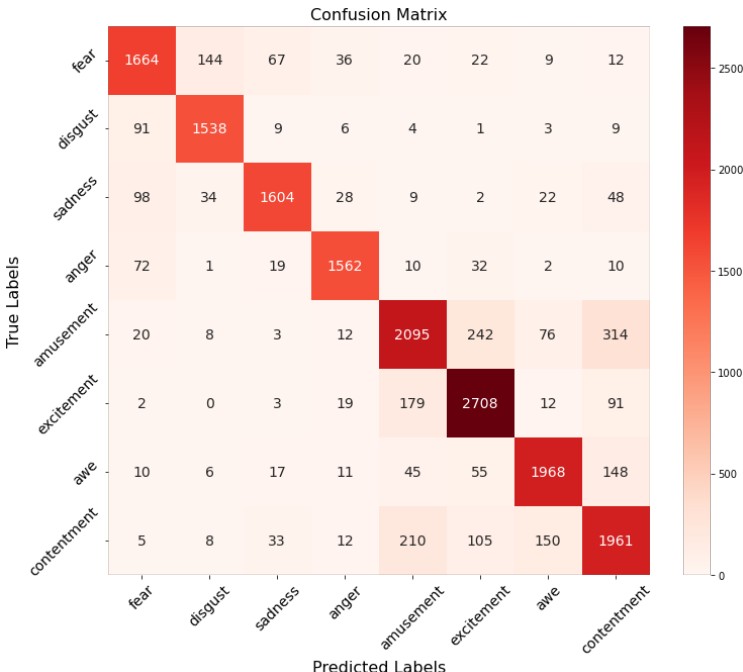

Figure A.3: ***Visual Emotion Analysis Confusion Matrix.*** *PerceptCLIP's emotion classification performance on the EmoSet test set across 8 emotion categories. Rows represent ground-truth labels; columns represent predicted labels.*

**Ground-Truth:** Sadness   **Predicted:** Contentment

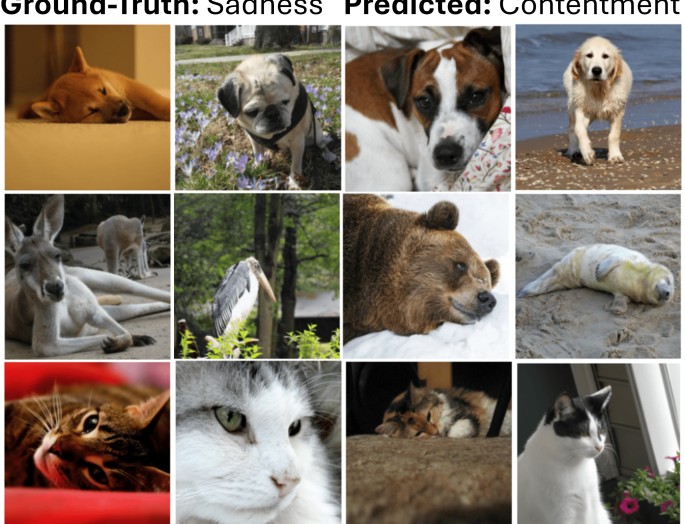

Figure A.4: ***Examples of Sadness Misclassified as Contentment.*** *Images where the ground-truth label is* sadness *but PerceptCLIP predicted* contentment*. Many of these examples feature animals, suggesting that the model may associate animal imagery with positive affect, leading to cross-valence misclassification.*

**Ground-Truth:** Anger   **Predicted:** Excitement

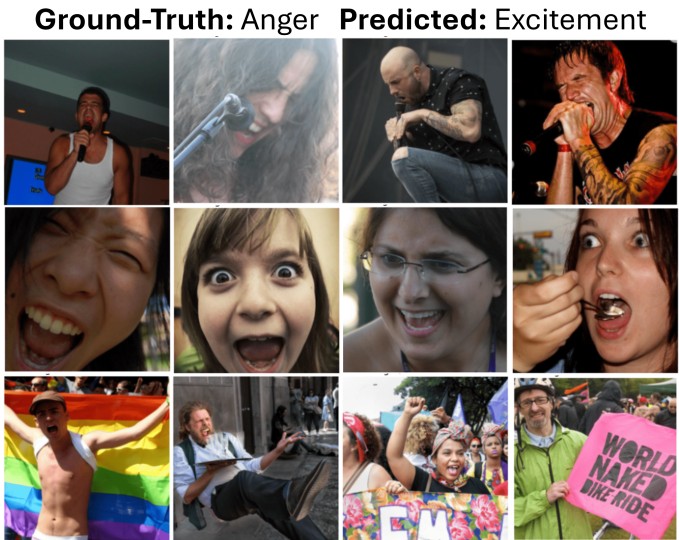

Figure A.5: ***Examples of Anger Misclassified as Excitement.*** *Images where the ground-truth label is* anger *but PerceptCLIP predicted* excitement*. Many of these examples depict people with open mouths, such as during protests or intense singing, suggesting a potential model bias toward interpreting such expressions as* excitement*. This highlights the subjectivity of emotion perception and the influence of content-specific visual cues.*

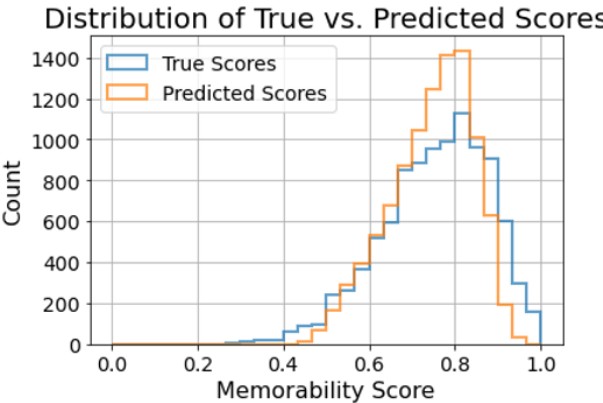

Figure A.6: ***Distribution of Ground-Truth vs. Predicted Memorability Scores.*** *Histogram comparing ground-truth and predicted memorability scores on the LaMem test set split1. The predicted scores are more concentrated around the center of the scale, indicating that the model tends to underestimate highly memorable images and overestimate less memorable ones.*

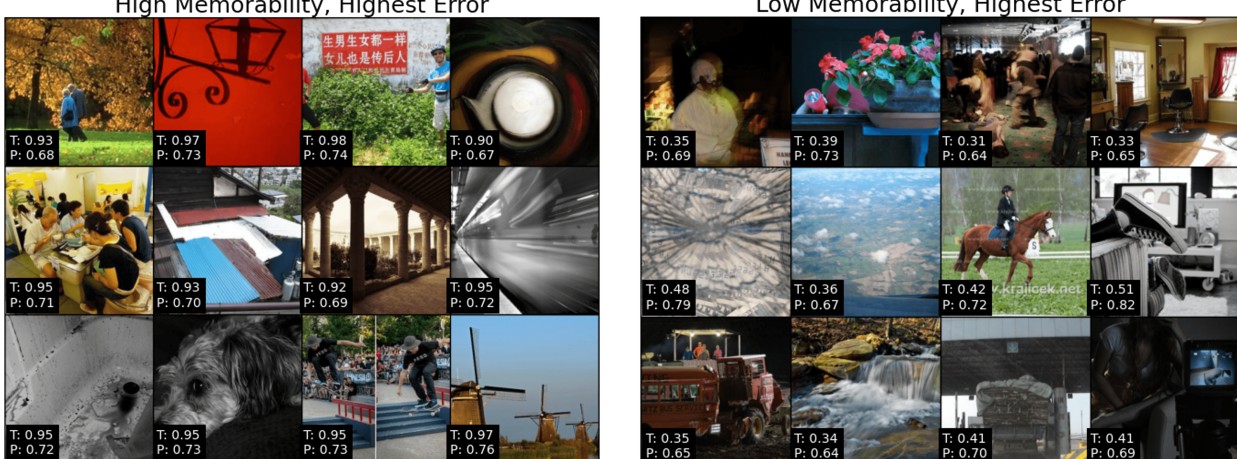

Figure A.7: ***Examples of Images with Extreme Prediction Errors.*** *Visualization of images from the LaMem test set with the largest prediction errors at both the low and high ends of the memorability spectrum. Each image is annotated with its ground-truth (T:) and predicted (P:) memorability scores.*

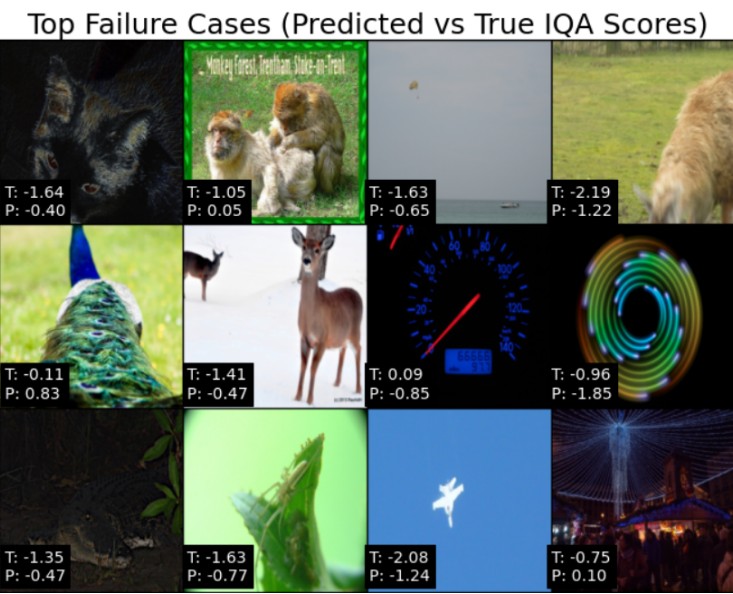

Figure A.8: **High-Error Examples in IQA.** *Visualization of images from the KonIQ10k test set with the largest differences between predicted and ground-truth quality scores. Both predicted and ground-truth scores were standardized using z-score normalization. Each image is annotated with its ground-truth (T:) and predicted (P:) quality scores.*

# B  Broader Impact Statement

This work introduces PerceptCLIP, a framework for visual perceptual tasks such as image memorability, image quality, and visual emotion analysis, built upon CLIP. While PerceptCLIP leverages CLIP's alignment with human judgment, it may inherit its pretraining data, which consists of large-scale internet content that can contain social and cultural biases. As a result, PerceptCLIP may reflect such biases in its predictions. Additionally, while our model predicts general perceptual responses based on crowd-sourced annotations, future extensions involving personalization or applications to personal content could raise ethical and privacy concerns.

