# OpenReview forum: "Don’t Judge Before You CLIP: A Unified Approach for Perceptual Tasks"
_TMLR — Accepted by TMLR_

### Review · Reviewer_2MdG · 2025-06-04

**Summary Of Contributions:**

This paper introduces a unified framework built on CLIP representations for visual perception tasks, which introduces tunable LORA and MLP layers.  The authors argue that CLIP can be repurposed for various perceptual tasks without requiring bespoke architectural changes.

Through comprehensive experiments, the proposed models are demonstrated to achieve state-of-the-art results on three distinct benchmarks: image memorability prediction, no-reference image quality assessment, and visual emotion analysis.

The authors demonstrate that joint training across heterogeneous datasets for the same task yields meaningful gains, especially when specific datasets are small. This empirical finding reinforces the value of multi-dataset learning and suggests practical strategies for alleviating data scarcity in perceptual modeling.

**Audience:**

No

**Claims And Evidence:**

Yes

**Requested Changes:**

Clarify the difference with other works, novelty, and technical contributions.

**Strengths And Weaknesses:**

Strength
- The proposed method is technically sound, simple yet effective
- The experiments are comprehensive, verifying the effectiveness of three tasks.
- Some experiments, such as dataset scaling, can provide some insight

Weakness
- The proposed method lacks technical contribution and novelty, as CLIP has already been utilized in the studied tasks in this paper, including IQA [1], emotion classification [2], and Image Memorability Prediction [3]. So, the observation about using CLIP for perception tasks is not new, and the method design using LORA to tackle these tasks is also not new. The authors also need to compare with these state-of-the-art.

[1] Wang, Jianyi, Kelvin CK Chan, and Chen Change Loy. "Exploring clip for assessing the look and feel of images." Proceedings of the AAAI conference on artificial intelligence. Vol. 37. No. 2. 2023.

[2] Yao, Yanyin, et al. "VLM-EMO: Context-Aware Emotion Classification with CLIP." 2024 5th International Seminar on Artificial Intelligence, Networking and Information Technology (AINIT). IEEE, 2024.

[3] Martín-Fernández, Iván, et al. "Video memorability prediction from jointly-learnt semantic and visual features." Proceedings of the 20th international conference on content-based multimedia indexing. 2023.

---

> ### Author Response · Authors · 2025-06-16
>
> Thank you for your time and effort in reviewing our paper. We will clarify our contribution and the differences from other works.
>
> ***Our main contribution*** is in demonstrating that strong performance on visual perceptual tasks can be achieved without complex technical designs. While CLIP has been applied in various domains, it has not been explored within a simple, unified framework across multiple perceptual tasks, as we propose. Since our focus is on leveraging CLIP’s perceptual prior, we also emphasize the role of LoRA in preserving this prior while enabling effective adaptation—unlike prior works that either fully fine-tuned or completely froze the encoder. Altogether, our work presents a practical framework, thorough analysis, and strong empirical results that we believe will support and inspire future research in visual perceptual modeling.
>
> ***We will now address the three cited papers***, and clarify how our work differs from those previous works:
> * *[1] Wang et al.:* This is indeed a relevant work (also cited in our related work), which uses either frozen CLIP or prompt learning. While related, its performance on IQA benchmarks is notably lower—for example, SRCC of 0.895 on KonIQ and 0.864 on SPAQ, compared to 0.945 and 0.930 with our PerceptCLIP model. We presented in the paper tables only a subset of recent SOTA methods for clarity.
> * *[2] VLM-EMO:* We have now added a reference to this work in the revised related work section. This paper leverages CLIP’s vision-language capabilities, fully fine-tunes the visual encoder, and uses curated and learnable prompts. As we show, full fine-tuning can degrade performance by removing CLIP’s prior knowledge. Additionally, the task differs: VLM-EMO predicts the emotion of a person in the image, whereas we predict the emotion evoked by the image itself (and therefore use different datasets). To the best of our knowledge, no code or pretrained models are available, preventing direct comparison on our datasets. Having said that, we now trained a model using our framework on CAER-S (their main dataset) and achieved 90.2% accuracy, outperforming their reported 88.02%.
> * *[3] Martín-Fernández et al.:* Thank you for pointing to this paper, we have added this reference to the revised version of the paper. This paper focuses on video memorability, not image memorability, and does not provide code to evaluate on our datasets. In addition, they also fully fine-tuned the CLIP backbone.
>
> As we mentioned in our related work section, we acknowledge that other studies have used CLIP as a backbone; however, their focus and motivation differ from ours. In most of these works, CLIP was primarily used for text-visual interaction or as a generic transformer backbone, rather than being explicitly selected for its strong perceptual prior, which is central to our approach. Their design choices, such as using curated prompts, learnable text embeddings, or full fine-tuning, reflect this emphasis. Furthermore, our paper includes hypotheses on why CLIP is effective for perceptual tasks, evaluations across multiple tasks, fine-tuning ablations, and attention-shift visualizations. We believe that this depth of exploration, combined with our model’s superior performance, represents a meaningful contribution to the community.

---

### Review · Reviewer_rTfe · 2025-06-07

**Summary Of Contributions:**

This paper presents a novel unified approach called PerceptCLIP for several image perceptual tasks, namely image memorability prediction, no-reference image quality assessment, and visual emotion analysis. The proposed approach leverages prior knowledge of the pre-trained CLIP model, which was indicated to have a good correlation with human judgement. To solve each of the perceptual tasks, the pre-trained CLIP model is fine-tuned using lightweight LoRA adapters on the weights of the attention layer (namely, key (K), query (Q), and value(V)), and the per-task trainable MLP head is added on top of CLIP embeddings, which outputs the final prediction to the task. Experimental results show that PerceptCLIP achieves state-of-the-art results on all the three tasks across multiple benchmark datasets, outperforming task-specific models that are often more complex and parameter-heavy. Moreover, the model demonstrates strong generalization, outperforming existing methods in cross-dataset evaluations, and benefits from joint training on multiple datasets from the same task. This suggests that the proposed model learns robust perceptual features rather than overfitting to individual datasets. Finally, the paper presents rich ablation study, showing benefits of utilising CLIP over other visual encoders and justifying the proposed tuning setup.
To sum up, PerceptCLIP proposes a unified framework for several perceptual tasks, that shows state-of-the-art results with good generalisation across multiple tasks and datasets, and benefits from universality of the architecture across these tasks, efficient training,  and interpretability of the learned features.

**Audience:**

Yes

**Broader Impact Concerns:**

-

**Claims And Evidence:**

Yes

**Requested Changes:**

I would only strongly request ablation on LoRA hyperparameters, i.e. r and α, at least for one (which is used in main experiments) version of CLIP backbone. Other things mentioned in the "weaknesses" section would be beneficial, but not crucial in my opinion.

**Strengths And Weaknesses:**

I like the paper in general, and I find the proposed approach simple and concise, yet quite powerful, which is a strong mixture in my opinion. Here's the list of strengths:
- The proposed approach is universal across several tasks (up to the MLP head, which is task-specific);
- PerceptCLIP utilises CLIP's pre-trained knowledge with little added trainable parameters, which makes it efficient compared to many existing approaches;
- Three considered tasks are different in nature, e.g. image memorability prediction should be related to perceiving semantic features of the image, image quality assessment mostly related to the fine-grained non-semantic details of the image, and visual emotion analysis is related to the specific information on the image, i.e. emotions;
- Extensive experiments show great performance across multiple datasets on all the three tasks and good generalisation across datasets. This suggests that the model does learn robust perceptual features rather than overfitting to individual datasets.
- Good ablation study. Essential cases of using fully-frozen CLIP, fully fine-tuned CLIP and fine-tuned CLIP using proposed LoRA approach are considered, which clearly shows the advantage of the proposed LoRA-based training. Ablation on the choice of the backbone model supports the choice of the CLIP, and different version of the CLIP backbone are considered. Ablation on attention heads interpretability is really nice, as it shows that the model did actually learn meaningful features that we would like it to learn, and it's performance is likely connected to the learned features, and not some other hidden reason.

Weaknesses (all except the first one is more like "what would be nice to have"):
- The paper has a good ablation on many factors, but the ablation on LoRA hyperparameters is missing. The authors use r=16 and α=8, but it would be beneficial to see how robust the model is to these parameters, and, maybe, what differences in attention maps it exhibits based on the values of these parameters. Moreover, there is an ablation on the choice of backbone (MAE, DINOv2, and CLIP), and though  they are all using ViT-L/14, different choices of r and α might be the best for different backbones.
- There're some more modern CLIP-like architectures, namely SigLIP, SigLIP2, TULIP. Would be interesting to see if their prior knowledge is better for this task
- I am a little concerned about CLIP having some perceptual biases, e.g. biased judgement on emotions of different people. It would be beneficial to explore that, although I am not sure how important that is for such kind of papers.
- There is only one dataset used for computing metrics on the task of image memorability. I am not sure if there exist more, but in case if not, it might be good to perform a user study to show generalisability of the method.

---

> ### Author Response · Authors · 2025-06-16
>
> Thank you for your time and effort in reviewing our paper and for your beneficial suggestions. We have conducted new experiments where possible to address your points. The results were added either to the main paper or to the appendix in the revised version.
>
> ***Ablation on LoRA hyperparameters:*** During our work, we did not observe a significant influence of the LoRA r and α parameters on performance—might be because LoRA already updates only a portion of the weights. Following your suggestion, we have now systematically experimented with different values of r and α on our main model and on KonIQ10k dataset from the IQA task ( see appendix A10 and A11). As shown in the table below (SRCC results), these parameters have only a minimal impact on model performance. We expect this low sensitivity to the parameters to be general, and therefore likely to apply other backbones as well.
> | r / α   | α = 4 | α = 8 | α = 16 |
> |:--------|:------:|:------:|:-------:|
> | **r = 8**   | 0.945  | 0.944  | 0.944   |
> | **r = 16**  | 0.944  | 0.945  | 0.945   |
> | **r = 32**  | 0.945  | 0.944  | 0.944   |
>
> ***Modern CLIP-like architectures:*** Following your suggestion, we experimented with training the TULIP model on one dataset from each task and did not observe significant improvement compared to CLIP (see table below and Table A8 in the revised appendix). While other architectures like SigLIP or SigLIP2—and further adaptation—may yield different results, we leave a more comprehensive comparison for future work.
> | | **IQA**  |           | **VEA**  | **Memorability** |         |
> |--------------|---------------------|-----------|--------------------|---------------------------|---------|
> |              | **(KonIQ)**           | | **(EmoSet-8)**           | **(LaMem)**                 |  |
> |              | **SRCC↑**           | **PLCC↑** | **ACC↑**           | **SRCC↑**                 | **MSE↓** |
> | **MAE**      | 0.924               | 0.935     | 0.7681             | 0.699                     | 0.0078  |
> | **DINOv2**   | 0.937               | 0.950     | 0.8252             | 0.729                     | 0.0071  |
> | **TULIP**    | 0.942               | 0.950     | 0.8410             | 0.682                     | 0.0076  |
> | **CLIP**     | 0.945               | 0.954     | 0.8528             | 0.741                     | 0.0069  |
>
> ***CLIP Perceptual Biases:*** We agree that this is a relevant and interesting concern. However, verifying such perceptual biases—especially across subjective dimensions like emotion—is a non-trivial task that would require carefully designed studies. We see this as an important direction for future exploration.
> Image memorability datasets: To the best of our knowledge, LaMem remains the most significant dataset for evaluating image memorability. Two smaller datasets—SUN and THINGS—do exist, but very few methods have reported results on them, and most lack available code. That said, we did include THINGS in our multi-dataset training (Table 5) and in the generalization results in Appendix Table A1. Based on your suggestion, we have now also evaluated our model on SUN (see table below and revised appendix Table A2), comparing it to the few available results from other methods. This will be added to the appendix in the revised version.
> |      **SUN** Dataset   |          |        |
> |-----------------------|-----------------|--------|
> |                       | **SRCC↑**       | **MSE↓** |
> | **MemNet**                | 0.610           | unknown |
> | **AMNet**                | 0.649           | 0.011 |
> | **PerceptCLIP (Ours)**| 0.708       | 0.014   |

---

### Review · Reviewer_vAhY · 2025-06-15

**Summary Of Contributions:**

This paper proposes a simple unified architecture that leverages the pre-trained CLIP model as a foundation for various visual perceptual tasks like Image Memorability Prediction, No-Reference Image Quality Assessment, and Visual Emotion Analysis. Across all three tasks, the proposed approach reportedly achieves SOTA results, outperforming specialized prior methods on standard benchmarks. An important claimed benefit is improved generalization: because perceptual data is scarce and often overfits, the authors test cross-dataset transfer and report substantially higher correlations than past models.

**Audience:**

Yes

**Broader Impact Concerns:**

CLIP was trained on a vast amount of internet data, which is known to include various social and cultural biases. If PerceptCLIP inherits these biases, its predictions of human judgments may be skewed or unfair for certain groups.

Visual emotion analysis in particular can be sensitive. If a model can predict emotional reaction to images, it might be one step away from trying to infer a viewer’s emotional state from their images or social media content.

**Claims And Evidence:**

Yes

**Requested Changes:**

The authors should explicitly clarify in the introduction and title what “unified” entails in this work. It would help to state that the framework is unified at the architectural level but that separate fine-tuning is done per task.

To give a more nuanced picture, the authors should add some analysis of where PerceptCLIP falls short. For instance, are there certain kinds of images where the model’s predictions disagree with human judgments?

**Strengths And Weaknesses:**

[Strengths]
The paper presents a single, general solution for multiple perceptual tasks by re-using the same CLIP-based model structure with minimal changes. The use of LoRA to fine-tune only CLIP’s attention layers means very few parameters are learned for each task.

The proposed method achieves strong results that meet or exceed SOTA on all three evaluated tasks. These performance gains, combined with the cross-dataset generalization tests, demonstrate that the CLIP-based solution is not only competitive but more robust across data variations than specialized models.

The model does not rely on any task-specific feature engineering, auxiliary networks, or human-provided insights. The paper’s clarity in describing this unified framework is good, and the rationale is intuitively well-motivated.

[Weaknesses]
The central idea of leveraging a large pre-trained model (CLIP) for downstream tasks with lightweight fine-tuning is incremental and not particularly novel. LoRA and related parameter-efficient fine-tuning techniques are established methods, and the decision to tune only attention layers is more an implementation detail than a deep insight.

The paper’s title and narrative emphasize a unified approach for perceptual tasks, implying a general solution across many tasks. However, the experimental validation is limited to three tasks, and the approach treats each task separately with separate fine-tuning and heads. As a result, the claim of a general unified solution feels somewhat overstated.

---

> ### Author Response · Authors · 2025-06-16
>
> Thank you for your time and effort in reviewing our paper. We have conducted additional analyses on PerceptCLIP’s failure cases and confusion patterns to gain deeper insight, and we address the other points raised in your review below. These results, along with the revised text, have already been incorporated into the updated version of the paper.
>
> ***Clarification of the term “Unified”:*** Thank you for highlighting this point. We have now clarified or replaced the use of the term “unified” in both the abstract and the introduction with more accurate wording (see edits below). Regarding the title, we use the term “unified approach” in the general sense of applying an identical architectural framework across perceptual tasks. We hope this distinction is clearer in the revised version, but we are happy to make further adjustments if needed.
>
> *Modifications* -  Abstract: “We propose a unified architectural framework for solving multiple different perceptual tasks…”  → “We propose an identical architectural framework for solving multiple different perceptual tasks…” Introduction:  “In this paper we propose a unified architectural framework, named PerceptCLIP, for solving multiple different perceptual tasks.”  → “In this paper we propose a unified approach, named PerceptCLIP, for solving multiple different perceptual tasks using identical architectural frameworks.”
>
>
> ***PerceptCLIP Failure Analysis:*** We conducted additional analysis experiments following the reviewer’s suggestion (see Appendix Section A4). For Memorability Prediction, we observed that PerceptCLIP’s predictions tend to regress toward the mean (Figure A6), and the failure cases show no clear visual trend (Figure A7). For Image Quality Assessment (IQA), we observed only subtle patterns in failure cases (Figure A8), possibly involving complex images that require high-level understanding. For Visual Emotion Analysis, a confusion matrix (Figure A3) shows that most misclassifications occur between semantically similar emotions (e.g., amusement vs. contentment, fear vs. disgust), which are often hard to distinguish even for annotators. In less similar confusions (i.e., sadness misclassified as contentment, Figure A4), many images contained animals, suggesting a potential bias linking animals to positive emotions. Another frequent confusion was anger vs. excitement (Figure A5), often in images of people with open mouths (e.g., protests or singing), highlighting the nuanced and subjective nature of emotional perception.
>
> While PerceptCLIP performs well overall, this analysis reveals areas for future improvement, including content-driven emotion biases and regression effects. We thank the reviewer for encouraging this deeper investigation.
>
>
> ***Contribution:*** Our main contribution lies in the combination of clear motivation, implementation aligned with that motivation, strong empirical results, and extensive analysis, rather than a specific novel architectural idea. We show that strong performance on visual perceptual tasks can be achieved without complex technical designs. Our work offers a practical and reproducible framework that we believe will benefit future research in perceptual modeling.

---

### Decision · Action_Editor_Tsjv · 2025-07-22

**Recommendation:** Accept as is

**Additional Comments:**

This paper presents a method that uses the CLIP model to solve several perceptual tasks. A nice fine-tuning method using LORA and MLP, and a careful implementation, make the solution impressive. The applicability for image memorability prediction, image quality assessment, and emotion analysis is validated. The results are impressive, and the approach works well across different datasets. The authors include useful ablation studies and interpretability analysis. Technical contribution is good; writing is clear, the method is practical, and claims are well supported by the experiments. I recommend acceptance.

**Audience:**

Yes

**Audience Explanation:**

The work will be of interest to readers working on perceptual modeling, vision-language systems, and representation learning.

**Claims And Evidence:**

Yes

**Claims Explanation:**

The paper is clearly written and technically sound. It includes solid  experiments, detailed analysis, and supports its main claims well.